# In vitro and in vivo characterization of SARS-CoV-2 resistance to ensitrelvir

Maki Kiso[1], Seiya Yamayoshi [1,2,3] ✉, Shun Iida [4], Yuri Furusawa[1,3], Yuichiro Hirata[4], Ryuta Uraki [1,3], Masaki Imai [1,2,3], Tadaki Suzuki [4] & Yoshihiro Kawaoka [1,3,5,6] ✉

Ensitrelvir, an oral antiviral agent that targets a SARS-CoV-2 main protease (3CLpro or Nsp5), is clinically useful against SARS-CoV-2 including its omicron variants. Since most omicron subvariants have reduced sensitivity to most monoclonal antibody therapies, SARS-CoV-2 resistance to other antivirals including main protease inhibitors such as ensitrelvir is a major public health concern. Here, repeating passages of SARS-CoV-2 in the presence of ensitrelvir revealed that the M49L and E166A substitutions in Nsp5 are responsible for reduced sensitivity to ensitrelvir. Both substitutions reduced in vitro virus growth in the absence of ensitrelvir. The combination of the M49L and E166A substitutions allowed the virus to largely evade the suppressive effect of ensitrelvir in vitro. The virus possessing Nsp5-M49L showed similar pathogenicity to wild-type virus, whereas the virus possessing Nsp5-E166A or Nsp5-M49L/E166A slightly attenuated. Ensitrelvir treatment of hamsters infected with the virus possessing Nsp5-M49L/E166A was ineffective; however, nirmatrelvir or molnupiravir treatment was effective. Therefore, it is important to closely monitor the emergence of ensitrelvir-resistant SARS-CoV-2 variants to guide antiviral treatment selection.

Three years have passed since the emergence of severe acute respiratory syndrome coronavirus 2 (SARS-CoV-2). During the pandemic, many variants of concern (VOCs) appeared and caused several waves of infection. Among these variants, omicron (lineage B.1.1.529) was identified at the end of 2021 and became globally dominant[1]. The ongoing global epidemic has resulted in the accumulation of amino acid substitutions throughout the genome of the virus. Recent subvariants including XBB, XBB.1.5, BQ.1.1, and CH.1.1 are less susceptible to therapeutic monoclonal antibodies against the spike protein of SARS-CoV-2[2–4]. In contrast, antiviral compounds against SARS-CoV-2 such as ensitrelvir (S-217622), nirmatrelvir (PF-07321332), and molnupiravir (EIDD-1931) remain effective against such omicron variants and their subvariants[2–9]. Ensitrelvir and nirmatrelvir interfere with 3CL protease [3CLpro; also known as main protease (Mpro) and nonstructural protein 5 (nsp5)], whereas remdesivir and molnupiravir target the viral RNA-dependent RNA polymerase. As these drugs are increasingly used in clinical settings, there is concern about the emergence of viruses with reduced sensitivity or resistance.

Ensitrelvir, which was discovered through a structure-based drug design strategy[10], has shown antiviral activity in cell culture, mice, and hamsters[11–13]. It has also been shown to prevent direct transmission from infected hamsters to naive hamsters[12]. In humans, ensitrelvir is well-tolerated and has favorable pharmacokinetics, including a long half-life[14]. In the Phase 2/3 study, ensitrelvir treatment of patients with mild to moderate COVID-19 resulted in rapid viral clearance[15,16]. When taken within 3 days of onset, ensitrelvir reduced the time to resolution

[1]Division of Virology, Institute of Medical Science, University of Tokyo, Tokyo, Japan. [2]International Research Center for Infectious Diseases, Institute of Medical Science, University of Tokyo, Tokyo, Japan. [3]The Research Center for Global Viral Diseases, National Center for Global Health and Medicine Research Institute, Tokyo, Japan. [4]Department of Pathology, National Institute of Infectious Diseases, Tokyo, Japan. [5]The University of Tokyo Pandemic Preparedness, Infection and Advanced Research Center, Tokyo, Japan. [6]Department of Pathobiological Sciences, School of Veterinary Medicine, University of Wisconsin–Madison, Madison, WI, USA. ✉e-mail: yamayo@ims.u-tokyo.ac.jp; yoshihiro.kawaoka@wisc.edu

of symptoms such as fever, runny nose, sore throat, cough, and fatigue by 24 h (from an average of 8 days to an average of 7 days)[17]. Now, ensitrelvir has received emergency regulatory approval from the Ministry of Health, Labour and Welfare of Japan and Fast Track designation from the U.S. Food and Drug Administration for the treatment of COVID-19.

Amino acid substitutions in nsp5 associated with resistance to nirmatrelvir have been reported[18–24]. Similarly, mutations that afford resistance to ensitrelvir have been found in Nsp5: M49I/L, S144A, E166A/V, L167F, and P168del reduce the potency of ensitrelvir in vitro[25]. In addition, naturally occurring amino acid changes such as M49I/L, S144A, P168del, and A173V contribute to reduced sensitivity to ensitrelvir in vitro[26]. A mutant virus possessing the P168del mutation showed similar replication kinetics as wild-type virus in VeroE6 cells, whereas a mutant virus possessing the A173V substitution was replication defective[26]. The M49I, G143S, and R188S substitutions have also been linked to ensitrelvir resistance[27]. Screening using a VSV-based system showed that the L167F substitution confers resistance to both nirmatrelvir and ensitrelvir[20]. A mutant virus harboring the L167F substitution replicated more slowly than wild-type virus and produced smaller plaques in VeroE6 cells[20]. The triple substitution of L50F, E166A, and L167F, which was selected by passaging wild-type virus in the presence of an nsp5 inhibitor, confers resistance to both nirmatrelvir and ensitrelvir[28]. A mutant virus carrying the L50F, E166A, and L167F substitutions showed similar pathogenicity in hamsters and transmitted to co-housed hamsters[29]. The Pharmaceuticals and Medical Devices Agency (PMDA) of Japan, reported that the D48G, M49L, P52S, S144A, and M49L/S144A substitutions confer reduced sensitivity to ensitrelvir in VeroE6/TMPRSS2 cells (https://www.pmda.go.jp/files/000249828.pdf). A crystal structure of the M49I mutant revealed that the greater hydrophobicity and steric volume of the isoleucine residue compared to the methionine residue shifted the ligand orientation within the binding site[27]. The M49I substitution also caused the loss of hydrogen bonds between ensitrelvir and T26, S144, and C145[27]. These alterations reduce virus sensitivity to ensitrelvir. Although the amino acid changes mentioned above have been associated with resistance to ensitrelvir, mutant viruses possessing mutations at positions 49 and 166 have not been well characterized in vitro or in vivo; growth comparisons in the presence or absence of ensitrelvir in vitro, detailed pathological comparisons, and studies comparing the efficacy of ensitrelvir nirmatrelvir, and molnupiravir in hamsters have not been done.

Accordingly, here, we attempted to obtain viruses with reduced sensitivity to ensitrelvir by passaging wild-type SARS-CoV-2 in vitro in the presence of ensitrelvir. With the knowledge of the amino acid substitutions in Nsp5 involved in resistance, we generated mutant viruses possessing these substitutions by using reverse genetics. We characterized the in vitro replication capability and pathogenicity in hamsters of these mutant viruses. We also evaluated the efficacy of antiviral treatments against these mutant viruses by using a hamster model.

## Results

### Selection of SARS-CoV-2 resistance to ensitrelvir

We attempted to determine whether a virus resistant to ensitrelvir (S-217622) would emerge upon serial passage. The wild-type delta variant was passaged twice in the presence of 0.25 μM ensitrelvir, twice at 0.5 μM, and then once at 1 or 5 μM. After a total of five passages, 6 clones (clones 11–16) from the samples at the 1 μM condition and 6 clones (clones 21–26) from the samples at the 5 μM condition were obtained by plaque purification and their nsp5 nucleotide sequences were analyzed by Sanger sequencing. Clones 11, 12, 13, 15, and 16 possessed the M49L substitution in nsp5, clones 14, 22, 23, 24, 25, and 26 possessed the E166A substitution in nsp5, and clone 21 possessed both substitutions. These substitutions have the potential to confer resistance to ensitrelvir. To confirm this, we generated mutant viruses possessing the M49L, E166A, or M49L and E166A substitutions in nsp5 (Nsp5-M49L, Nsp5-E166A, or Nsp5-M49L/E166A) based on the wild-type delta variant by using a BAC-based reverse genetics system[30]. The viruses were tested for sensitivity to ensitrelvir (S-217622) together with another nsp5 inhibitor, nirmatrelvir (PF-07321332), and the antiviral ribonucleoside analogs molnupiravir (EIDD-1931) and remdesivir (GS-441524). The $IC_{50}$ value of ensitrelvir (S-217622) against wild-type virus was 0.19 μM [95% confidence interval (CI), 0.16 to 0.24], whereas the $IC_{50}$ values against Nsp5-M49L, Nsp5-E166A, and Nsp5-M49L/E166A were increased to 11.6 μM (95% CI, 10.4 to 13.0), 1.72 μM (95% CI, 1.59 to 1.86), and 37.4 μM (95% CI, 32.5 to 43.2), respectively (Table 1). The sensitivity of Nsp5-M49L to nirmatrelvir (PF-07321332) was slightly increased, but that of the remaining two mutant viruses was not affected. Since we did not use a P-glycoprotein inhibitor, which inhibits the extreme efflux of nirmatrelvir from cells, with nirmatrelvir[31], the $IC_{50}$ values of nirmatrelvir were relatively high[32]. Molnupiravir (EIDD-1931) and remdesivir (GS-441524) showed a slightly higher $IC_{50}$ value against the three mutant viruses than against the wild-type virus, for reasons that remain unclear. These results indicate that Nsp5-M49L, Nsp5-E166A, and Nsp5-M49L/E166A show moderate, slight, and considerable resistance to ensitrelvir, respectively.

### Propagation of mutant viruses in the presence or absence of ensitrelvir in vitro

We next evaluated the growth of Nsp5-M49L, Nsp5-E166A, and Nsp5-M49L/E166A in vitro. VeroE6/TMPRSS2 cells were infected with each virus at an MOI of 0.001 and virus titers were determined at the indicated timepoints after incubation in the presence (dashed lines) or absence (solid lines) of 20 μM ensitrelvir (Fig. 1). Wild-type virus replicated efficiently in VeroE6/TMPRSS2 cells, reaching $10^6$ PFU/ml, whereas 20 μM ensitrelvir completely inhibited wild-type virus replication (Fig. 1a). In the absence of ensitrelvir, the growth of Nsp5-M49L and Nsp5-E166A was slightly delayed compared with that of wild-type virus (Fig. 1b, c), while Nsp5-M49L/E166A showed moderate growth delay (Fig. 1d). In the presence of ensitrelvir, propagation of Nsp5-M49L was severely impaired (Fig. 1b), Nsp5-E166A propagation was not detected at all (Fig. 1c), whereas growth of Nsp5-M49L/E166A was less affected (Fig. 1d). These results indicate that the M49L substitution in Nsp5 is responsible for resistance to ensitrelvir and the additional E166A substitution improves replication in the presence of ensitrelvir. Both M49L and E166A substitutions decrease the efficiency of virus propagation.

### Pathogenicity of mutant viruses in hamsters

We compared the pathogenicity of Nsp5-M49L, Nsp5-E166A, and Nsp5-M49L/E166A with that of wild-type virus in the hamster infection model. Hamsters were infected with $10^5$ PFU of each virus and body weight, respiratory functions, and virus titers in the nasal turbinates

**Table 1 | Sensitivity of mutant viruses to antivirals against SARS-CoV-2**

| Antiviral | Wild type | Nsp5-M49L | Nsp5-E166A | Nsp5-M49L/E166A |
|---|---|---|---|---|
| Ensitrelvir (S-217622) | 0.19 (0.16–0.24) | 11.6 (10.4–13.0) | 1.72 (1.59–1.86) | 37.4 (32.5–43.2) |
| Nirmatrelvir (PF-07321332) | 8.63 (7.97–9.32) | 3.26 (2.98–3.56) | 11.2 (8.24–15.0) | 8.19 (6.64–9.99) |
| Molnupiravir (EIDD-1931) | 3.31 (2.31–4.72) | 13.1 (11.8–14.5) | 13.5 (11.6–15.7) | 10.21 (8.94–11.7) |
| Remdesivir (GS-441524) | 0.83 (0.70–0.98) | 2.64 (2.48–2.81) | 2.58 (2.42–2.75) | 2.88 (2.66–3.11) |

$IC_{50}$ values (μM) with 95% confidence interval to antivirals were measured by performing a focus reduction assay using VeroE6-TMPRSS2-T2A-ACE2 cells.

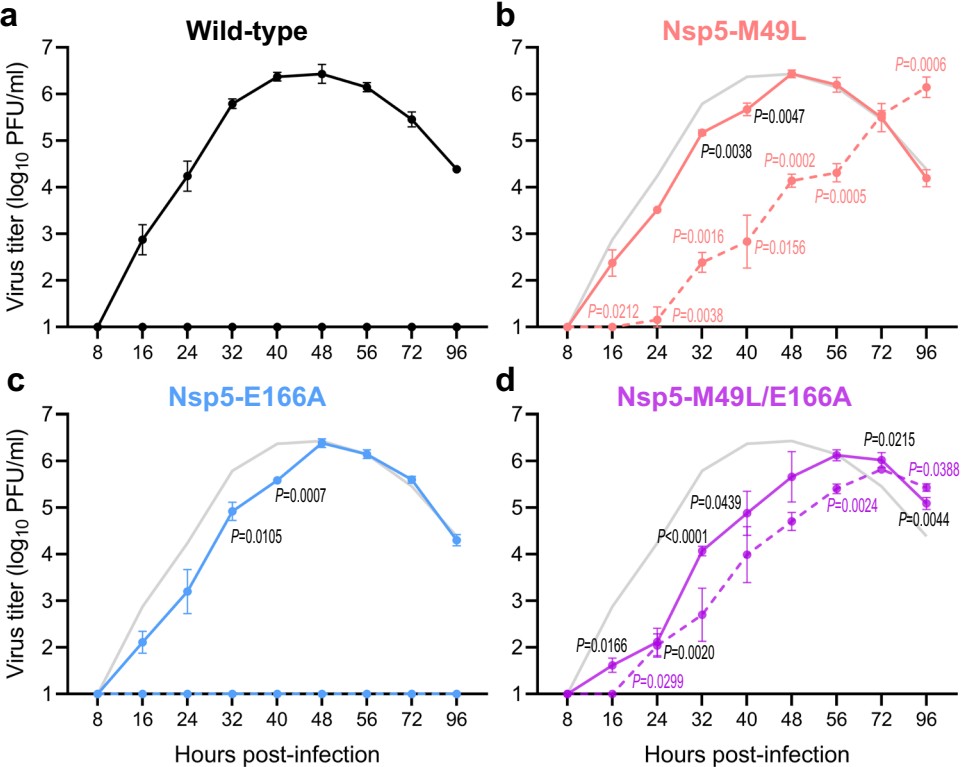

**Fig. 1 | Growth kinetics of Nsp5-M49L, Nsp5-E166A, and Nsp5-M49L/E166A in vitro.** The indicated viruses were inoculated into VeroE6/TMPRSS2 cells at an MOI of 0.001. After 8, 16, 24, 32, 40, 48, 56, 72, and 96 h incubation in the absence (solid line) or presence (dashed line) of 20 μM ensitrelvir, virus titers were determined by use of plaque assays on VeroE6/TMPRSS2 cells. Gray lines in **b**–**d** indicate the growth of wild-type virus without ensitrelvir shown in (**a**). The data shown are mean virus titers ± standard deviation ($n = 3$ independent experiments). Data were analyzed by using a two-way ANOVA followed by Dunnett's multiple comparisons. Black *P* values denote comparisons of each mutant virus without ensitrelvir to wild-type virus without ensitrelvir and colored *P* values denote comparisons of each mutant virus with ensitrelvir to each mutant virus without ensitrelvir.

and lungs were measured at each timepoint. Mock-infected hamsters gradually gained body weight, whereas hamsters infected with wild-type virus lost body weight until 7 dpi (Fig. 2a). Nsp5-M49L-infected hamsters showed similar body weight change to hamsters infected with wild-type virus, whereas those infected with Nsp5-E166A or Nsp5-M49L/E166A showed mild body weight loss compared with those infected with wild-type virus.

The respiratory function of the hamsters was assessed by measuring Penh and Rpef, which are surrogate markers for bronchoconstriction and airway obstruction, respectively. Wild-type virus significantly impaired Penh and Rpef at 5 dpi, as did Nsp5-M49L (Fig. 2b). Nsp5-E166A and Nsp5-M49L/E166A caused slight impairment of respiratory functions.

Virus titers in the nasal turbinates and lungs of infected hamsters were measured at 3 and 6 dpi. At 3 dpi, virus titers in the nasal turbinates were comparable for all viruses tested, whereas those in the lungs of hamsters infected with Nsp5-E166A or Nsp5-M49L/E166A were significantly lower than those of wild-type virus (Fig. 2c) At 6 dpi, there was no significant difference in virus titers in the nasal turbinates and lungs of the mutant virus-infected groups compared with the wild-type virus-infected group. To exclude the possibility of reversion or compensatory mutations in nsp5, we performed deep sequencing of the nasal turbinates and lungs from the infected hamsters at 3 and 6 dpi. Additional amino acid substitutions or reversions were not detected in most samples, although one nasal turbinate sample from a hamster infected with Nsp5-M49L/E166A at 6 dpi contained the E47K substitution in nsp5 at 17.1%. The viral titer of this sample was 5.49 ($\log_{10}$PFU/g), whereas the others were 5.14, 5.23, 4.62, and 5.23. Therefore, the E47K substitution does not appear to have a large effect on viral replication efficiency. Nevertheless, further analysis will be

required to characterize this substitution because it is located close to the enzyme active site of nsp5.

The lungs of hamsters infected with each virus were histopathologically analyzed at 3 and 6 dpi (Fig. 3a). H&E staining for all tested groups revealed infiltration of inflammatory cells such as mononuclear cells and neutrophils into the bronchi/bronchioles and no obvious inflammation in the alveoli at 3 dpi. At 6 dpi, bronchial and bronchiolar inflammation was improved compared with at 3 dpi, whereas severe alveolar inflammation was observed in all four groups. According to the histopathological scores at 6 dpi (Fig. 3b), there was a slight decrease in the severity of inflammation in the lungs of hamsters infected with Nsp5-E166A. In contrast, hamsters infected with Nsp5-M49L/E166A showed significantly less severe inflammation in their lungs compared to those infected with the wild-type virus. Viral RNA and protein were diffusely detected on bronchial and bronchiolar epithelium cells in all tested groups, whereas the number of viral RNA- and protein-positive cells in the alveolar area was slightly lower for the Nsp5-E166A- and Nsp5-M49L/E166A-infected groups than for the wild-type-infected group at 3 dpi (Fig. 3a). At 6 dpi, almost no viral RNA- or protein-positive cells were observed in any group (Fig. 3a).

Taken together, these data demonstrate that Nsp5-M49L maintains virulence comparable to that of wild-type virus, whereas Nsp5-E166A and Nsp5-M49L/E166A are slightly attenuated.

### Antiviral treatment of hamsters infected with an ensitrelvir-resistant virus

We next evaluated the efficacy of antiviral treatment of hamsters against the ensitrelvir-resistant virus. Hamsters infected with $10^5$ PFU of wild-type virus or Nsp5-M49L/E166A were treated with ensitrelvir at 60 mg/kg twice daily, nirmatrelvir at 250 mg/kg twice daily, or

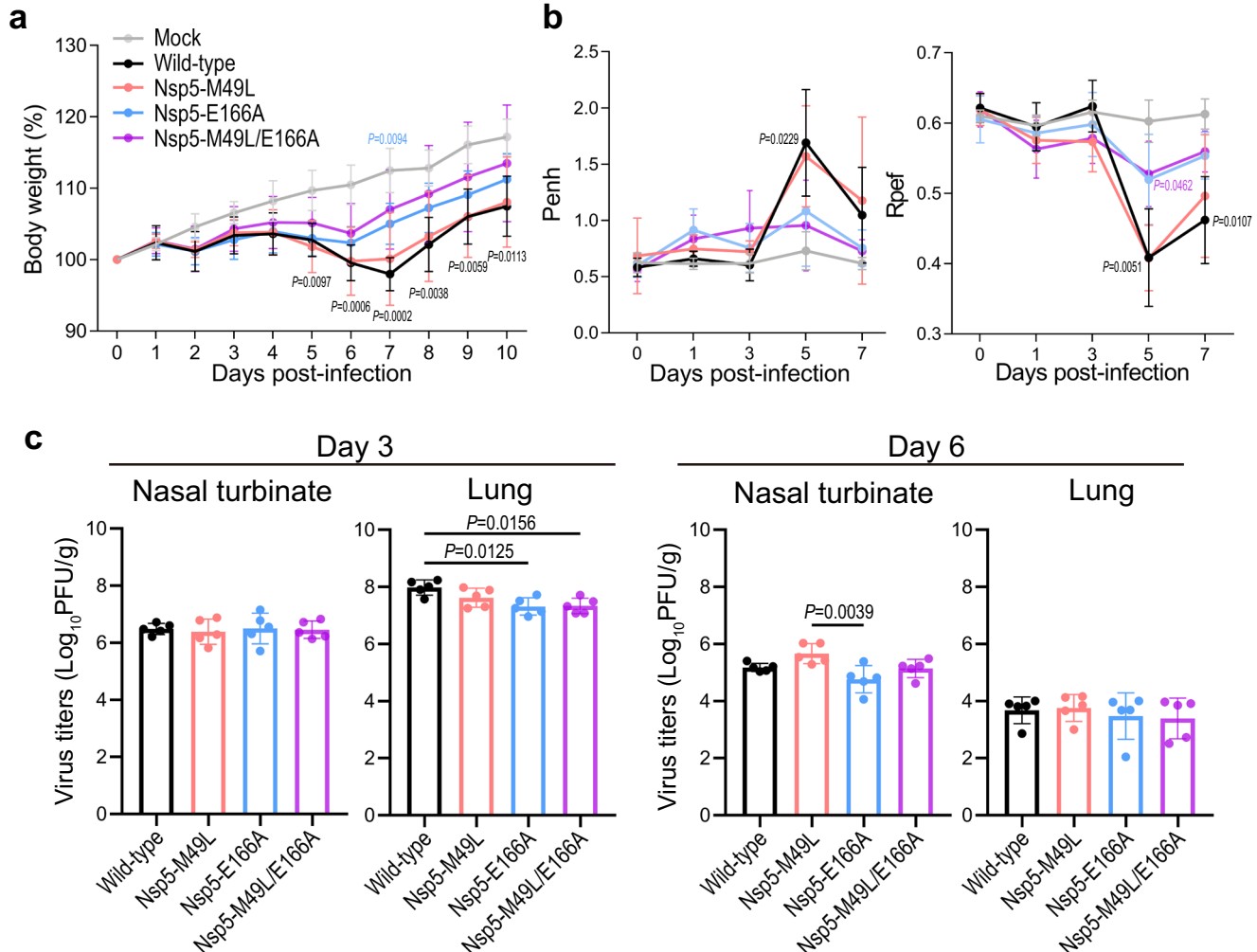

**Fig. 2 | Pathogenicity of Nsp5-M49L, Nsp5-E166A, and Nsp5-M49L/E166A in vivo. a, b** Hamsters were intranasally inoculated with $10^5$ PFU of the indicated virus or with phosphate-buffered saline (mock). Body weight (**a**) and respiratory functions (Penh and Rpef) (**b**) of virus-infected ($n = 5$) and mock-infected hamsters ($n = 5$) were measured daily for 10 days and by using whole-body plethysmography. Data are presented as the mean ± standard deviation and were analyzed by using a two-way ANOVA followed by Dunnett's multiple comparisons. (**c**) Virus propagation in the nasal turbinates and lungs of hamsters. Hamsters ($n = 10$) were intranasally inoculated with $10^5$ PFU of the indicated viruses and the nasal turbinates and lungs were collected at 3 and 6 dpi for virus titration ($n = 5$ per day). Virus titers were determined by use of plaque assays with VeroE6/TMPRSS2 cells. Points indicate data from individual hamsters and bars show the mean ± standard deviation. Data were analyzed by using a one-way ANOVA followed by Tukey's multiple comparisons.

molnupiravir at 250 mg/kg twice daily for three days beginning 1 dpi. Because nirmatrelvir was not co-administered with ritonavir, the efficacy of nirmatrelvir appears to be low. Virus titers in the nasal turbinates and lungs of the hamsters were measured at 4 dpi. In the hamsters infected with wild-type virus, ensitrelvir, nirmatrelvir, and molnupiravir significantly reduced the virus titers in both the nasal turbinates and lungs (Fig. 4, left panel); ensitrelvir dramatically suppressed virus replication, especially in the lungs. In the Nsp5-M49L/ E166A-infected hamsters, none of the three antivirals reduced the virus titers in the nasal turbinates; however, nirmatrelvir and molnupiravir significantly reduced the virus titers in the lungs but ensitrelvir did not (Fig. 4, right panel). These results show that treatment with nirmatrelvir or molnupiravir could be effective against ensitrelvir-resistant viruses in vivo.

## Discussion

Viruses resistant to anti-SARS-CoV-2 drugs are a major public health concern. Indeed, monoclonal antibody therapies face increasing escape mutants with amino acid substitutions in their spike protein. In contrast, antiviral compounds targeting virus proteins other than the spike continue to be efficacious and remain important countermeasures. However, amino acid substitutions that reduce sensitivity to such compounds have been found in viruses isolated from COVID-19 patients[18,19,33,34]. Here, passages of wild-type virus in the presence of ensitrelvir revealed that the M49L and E166A substitutions in nsp5 are associated with reduced sensitivity to ensitrelvir. Ensitrelvir resistance due to naturally occurring mutations and cross-resistance of nirmatrelvir-resistant mutations to ensitrelvir are caused by amino acid changes at positions 49 (M49I/L) and 166 (E166A/V)[25–27]. Therefore, these positions might be hot spots for amino acid substitutions that reduce sensitivity to ensitrelvir. Since recent omicron variants and subvariants naturally tend to obtain additional amino acid substitutions in nsp5 compared with the delta variant, amino acid substitutions at other positions might be selected during ensitrelvir treatment of patients. Therefore, we must monitor the amino acid substitutions in nsp5 that are detected in patients treated with ensitrelvir and examine the ensitrelvir sensitivity of isolates from these patients in laboratory experiments.

The M49L substitution reduced susceptibility to ensitrelvir and the in vitro growth of Nsp5-M49L was not completely suppressed by

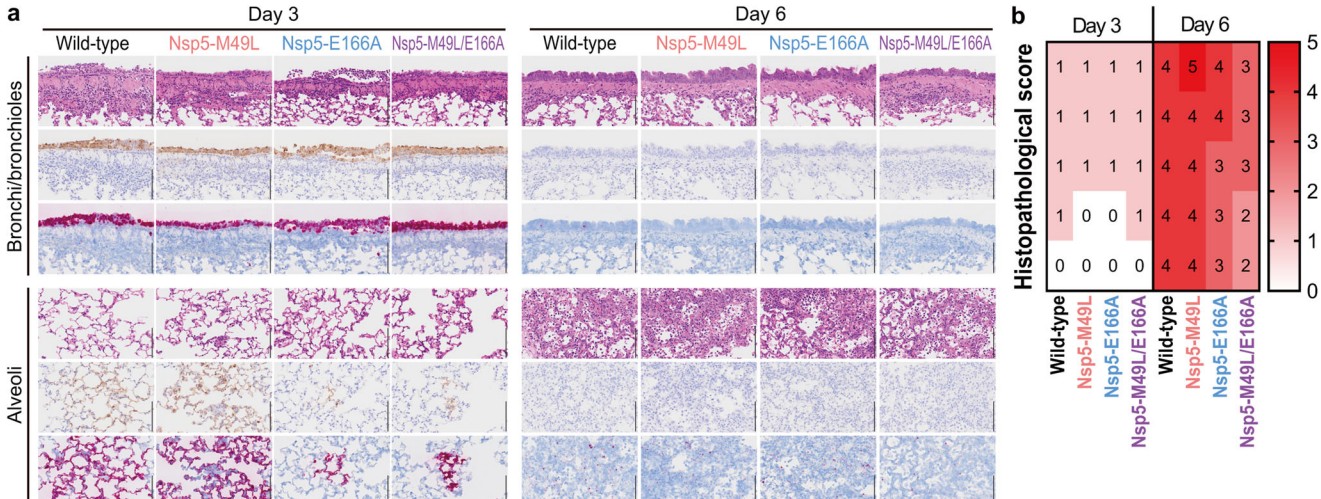

**Fig. 3 | Histopathological analysis of the lungs of infected hamsters. (a)** Hamsters were infected with $10^5$ PFU of the indicated virus and their lungs were collected at 3 or 6 dpi. Representative images of the bronchi/bronchioles and alveoli of hamsters (*n* = 5) are shown. Upper rows, H&E staining. Middle rows, immunohistochemistry detecting SARS-CoV-2 nucleocapsid protein. Lower rows, in situ hybridization targeting the nucleocapsid gene of SARS-CoV-2. Scale bars, 100 μm. **(b)** Histopathological scores of inflammation were determined based on the percentage of alveolar inflammation in a given area of a pulmonary section collected from each animal in each group by using the following scoring system: 0, no inflammation; 1, affected area (≤1%); 2, affected area (>1%, ≤10%); 3, affected area (>10%, ≤50%); and 4, affected area (>50%). An additional point was added when pulmonary edema and/or alveolar hemorrhage was observed. Therefore, histopathological scores of inflammations in the alveoli for individual animals ranged from 0 to 5.

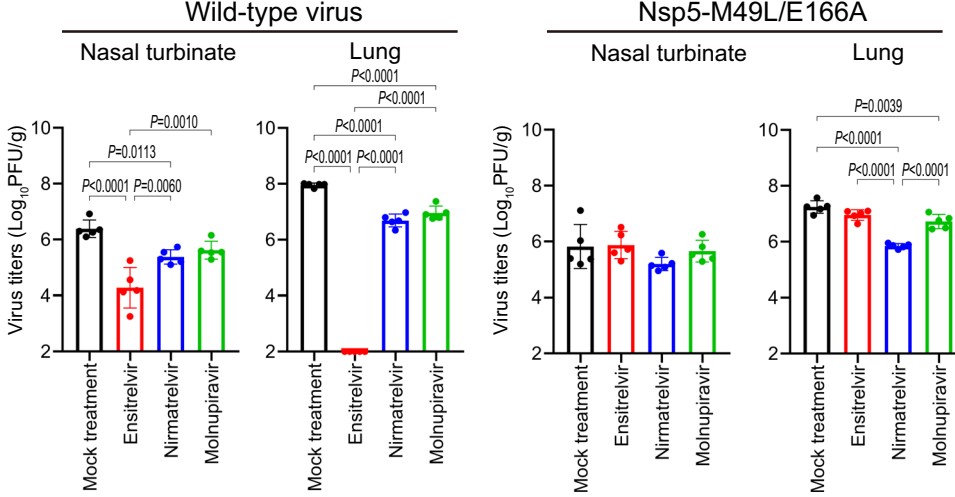

**Fig. 4 | Efficacy of three antivirals against wild-type virus and Nsp5-M49L/E166A in hamsters.** Wild-type virus or Nsp5-M49L/E166A was intranasally inoculated into hamsters (*n* = 20). At 1 dpi, hamsters were orally administered 0.5% methyl cellulose (mock treatment) (*n* = 5), ensitrelvir (*n* = 5), nirmatrelvir (*n* = 5), or molnupiravir (*n* = 5) for 3 days. Animals were euthanized at 4 dpi and their nasal turbinates and lungs were collected for the virus titration on VeroE6/TMPRSS2 cells. Points indicate data from individual hamsters and bars show the mean ± standard deviation. Data were analyzed by using a one-way ANOVA followed by Tukey's multiple comparisons.

ensitrelvir. The combination of the M49L and E166A substitutions further reduced susceptibility to ensitrelvir, resulting in similar growth kinetics of Nsp5-M49L/E166A in the presence and absence of ensitrelvir. Furthermore, ensitrelvir treatment did not reduce virus titers of Nsp5-M49L/E166A in the lungs of hamsters, although nirmatrelvir treatment was effective. These results demonstrate that Nsp5-M49L and Nsp5-M49L/E166A maintain viral fitness to a certain extent in vitro and in vivo, meaning that they have the potential to be a threat to public health as ensitrelvir-resistant viruses. Although amino acid substitutions at position 166, including E166D/G[18,19,35], E166V[21,22], and E166A[28], confer reduced sensitivity to nirmatrelvir, we confirmed that viruses with the E166A substitution, even in combination with the M49L substitution in the backbone of the delta variant, showed similar sensitivity to nirmatrelvir in vitro and in vivo. Since the efficacy of nirmatrelvir in this study was relatively low because a P-glycoprotein inhibitor or ritonavir was not co-administrated with nirmatrelvir[31,32], nirmatrelvir could be a treatment option against these resistant viruses. However, several amino acid substitutions such as E166V and L167F similarly confer resistance to both nirmatrelvir and ensitrelvir[25,28], indicating that both ensitrelvir and nirmatrelvir might fail to suppress the propagation of mutant viruses possessing such substitutions. Even so, molnupiravir and remdesivir likely remain as effective against such double-resistant viruses as against wild-type virus because the mechanism of action of both molnupiravir and remdesivir differs from that of the anti-nsp5 drugs such as ensitrelvir and nirmatrelvir. Further analyses

are required to determine whether a multi-drug resistant virus maintains viral fitness. To combat these multidrug-resistant viruses, we hope that many novel antivirals targeting other viral proteins will be approved for clinical use.

## Methods

### Ethics

All animal experiments were conducted in accordance with the University of Tokyo's Regulations for Animal Care and Use, which were approved by the Animal Experiment Committee of the Institute of Medical Science, the University of Tokyo. The committee acknowledged and accepted both the legal and ethical responsibility for the animals, as specified in the Fundamental Guidelines for Proper Conduct of Animal Experiment and Related Activities in Academic Research Institutions under the jurisdiction of the Ministry of Education, Culture, Sports, Science, and Technology of Japan.

### Biosafety statement

All experiments with SARS-CoV-2 were performed in enhanced biosafety level 3 containment laboratories at the University of Tokyo, which are approved for such use by the Ministry of Agriculture, Forestry, and Fisheries and the Ministry of Health, Labour and Welfare, Japan. All experiments were conducted by PhD-level scientists who are highly experienced in such studies. All individuals working with SARS-CoV-2 have been vaccinated multiple times with COVID-19 vaccines. Staff working in enhanced BSL-3 wear disposable overalls and powered air-purifying respirators. The enhanced BSL-3 facility at the University of Tokyo includes controlled access, effluent decontamination, negative air-pressure, double-door autoclaves, HEPA-filtered supply and exhaust air, and airtight dampers on ductwork connected to the animal cage isolators and biosafety cabinets. The structure is pressure-decay tested regularly. All personnel complete biosafety and BSL-3 training before participating in BSL-3-level experiments. Refresher training is scheduled on a regular basis. Virus inventory, secured behind two physical barriers, is checked regularly. Virus inventory is submitted once a year to the Ministry of Agriculture, Forestry, and Fisheries and the Ministry of Health, Labour and Welfare, Japan. Procedures in response to laboratory accidents are established.

### Cells

VeroE6/TMPRSS2 (JCRB 1819) cells were propagated in Dulbecco's modified Eagle's medium (DMEM) containing 10% fetal bovine serum (FBS), 1 mg/ml geneticin (G418; Invivogen), and 5 µg/ml plasmocin prophylactic (Invivogen). VeroE6-TMPRSS2-T2A-ACE2 cells were cultured in DMEM containing 10% FBS, 100 U/ml penicillin–streptomycin, and 10 µg/ml puromycin. HEK293T cells were cultured in DMEM supplemented with 10% FBS. All cells were maintained at 37 °C with 5% $CO_2$. The cells were regularly tested for mycoplasma contamination by using PCR and confirmed to be mycoplasma-free.

### Antivirals

Ensitrelvir (S-217622) was kindly provided by Shionogi Co., Ltd. Active components of remdesivir (GS-441524), molnupiravir (EIDD-1931), and nirmatrelvir (PF-07321332) were purchased from MedChemExpress. Compounds were dissolved in dimethyl sulfoxide for in vitro experiments or 0.5% methyl cellulose for in vivo experiments prior to use.

### Selection of SARS-CoV-2 resistance to ensitrelvir

SARS-CoV-2 delta variant (hCoV-19/USA/WI-UW-5250/2021) was sequentially passaged a total of five times in the presence of ensitrelvir at 0.25, 0.5, 1, or 5 µM in VeroE6/TMPRSS2 cells. In total, 12 clones were purified from the passaged viruses by plaque purification. The Nsp5 nucleotide sequence of each clone was determined by Sanger sequencing.

### Virus rescue using bacterial artificial chromosome (BAC)

The full-genome nucleotide sequence of SARS-CoV-2 delta variant (hCoV-19/USA/WI-UW-5250/2021) was assembled into the pBeloBAC11 vector to generate infectious cDNA clones under the control of a cytomegalovirus promoter as described previously[30]. The mutations responsible for the M49L, E166A, and M49L/E166A substitutions in nsp5 were introduced during the PCR step. To rescue these viruses, pBeloBAC11 encoding wild-type, Nsp5-M49L, Nsp5-E166A, or Nsp5-M49L/E166A was transfected into HEK293T cells. At 3 days post-transfection, the supernatant containing the viruses was collected and inoculated onto VeroE6/TMPRSS2 cells at 37 °C to prepare virus stocks. The stock viruses were deep sequenced to confirm the absence of unwanted mutations, and no position contained unwanted nucleotides that exceeded 10% of the population in all stock viruses.

### Deep sequence analysis

The whole genome of SARS-CoV-2 was amplified by using a modified ARTIC network protocol in which some primers were replaced or added[36]. In brief, viral RNA was extracted by using a QIAamp Viral RNA Mini Kit (QIAGEN). cDNA was synthesized by using a Lunar-Script RT SuperMix Kit (New England BioLabs) and subjected to a multiplexed PCR in two pools using ARTIC-N1 primers v5 and Q5 Hot Start DNA polymerase (New England BioLabs). The DNA libraries for Illumina NGS were prepared from pooled amplicons by using a QIAseq FX DNA Library Kit (QIAGEN) and then analyzed by using the iSeq100 System in 150-bp paired-end mode using an iSeq 100 i1 Reagent v2 (300-cycle) kit (Illumina). The reads were assembled by CLC Genomics Workbench (version 22, Qiagen).

### Focus reduction test

Antiviral susceptibilities were determined by using a focus reduction assay as previously reported[5–7]. Briefly, VeroE6-TMPRSS2-T2A-ACE2 cells in 96-well plates were infected with the indicated virus at 100–400 focus forming unit/well. After incubation at 37 °C for 1 h, the inoculum was replaced with 1% Methyl Cellulose 400 (FUJIFILM Wako Pure Chemical Corporation) in culture medium containing serial dilutions of the antiviral compounds. The cells were incubated for 18 h at 37 °C and then fixed with formalin. The cells were stained with a mouse monoclonal antibody against SARS-CoV-2 nucleoprotein, clone N45 (TAUNS Laboratories, Inc., Japan), followed by a horseradish peroxidase-labeled goat anti-mouse immunoglobulin (Jackson ImmunoResearch Laboratories Inc.). Foci were visualized by using TrueBlue Substrate (SeraCare Life Sciences). The focus numbers were quantified by using an ImmunoSpot S6 Analyzer, ImmunoCapture software, and BioSpot software (Cellular Technology). The 50% inhibitory concentration ($IC_{50}$) values and 95% confidence intervals were calculated by using GraphPad Prism 9.3.0 (GraphPad Software).

### Growth kinetics in vitro

VeroE6/TMPRSS2 cells were infected with the indicated virus at a multiplicity of infection (MOI) of 0.001. After incubation at 37 °C for 1 h, the inoculum was replaced with medium with or without 20 µM ensitrelvir. Cell culture supernatants were collected at 8, 16, 24, 32, 40, 48, 56, 72, and 96 h post-infection. Virus titers were determined by use of a plaque assay in VeroE6/TMPRSS2 cells.

### Experimental infection of Syrian hamsters

Five- to six-week-old male Syrian hamsters (Japan SLC) were used in this study. For the pathogenicity study, hamsters ($n = 5$) were intranasally inoculated with $10^5$ plaque forming unit (PFU) of the indicated virus. Body weights were measured daily before inoculation and for 10 days post-infection (dpi). Respiratory parameters [Penh (a non-specific assessment of breathing patterns) and Rpef (a measure of airway obstruction)] were also measured by using a whole-body plethysmography system (PrimeBioscience) as previously described[37,38].

For the virus titration and pathological analysis, hamsters ($n = 10$) were intranasally infected with $10^5$ PFU of the indicated virus. At 3 and 6 dpi, the animals ($n = 5$ per timepoint) were euthanized and nasal turbinates and right lungs were collected. The virus titers in these organs were determined by use of plaque assays on VeroE6/TMPRSS2 cells. The left lungs were fixed in 4% paraformaldehyde in PBS and processed for paraffin embedding. The paraffin blocks were cut into 3-μm-thick sections and mounted on silane-coated glass slides, and then haematoxylin and eosin (H&E) stained for histopathological examination. To detect SARS-CoV-2 RNA, in situ hybridization was performed using an RNA scope 2.5 HD Red Detection kit (Advanced Cell Diagnostics) with an antisense probe targeting the nucleocapsid gene of SARS-CoV-2 (Advanced Cell Diagnostics) as previously described[39]. Tissue sections were also processed for immunohistochemical staining with a rabbit monoclonal antibody for SARS-CoV nucleocapsid protein (Sino Biological), which cross-reacts with SARS-CoV-2 nucleocapsid protein. Specific antigen-antibody reactions were visualized by means of 3,3'-diaminobenzidine tetrahydrochloride staining using the Dako Envision system (Dako Cytomation). Histopathological scores of inflammation in the alveolar regions were determined based on the percentage of alveolar inflammation in a given area of a pulmonary section collected from each animal in each group by using the following scoring system[38]: 0, no inflammation; 1, affected area (≤1%); 2, affected area (>1%, ≤10%); 3, affected area (>10%, ≤50%); 4, affected area (>50%). An additional point was added when pulmonary edema and/or alveolar hemorrhage was observed. Therefore, histopathological scores of inflammation in the alveoli for individual animals ranged from 0 to 5.

For the treatment study, wild-type virus or Nsp5-M49L/E166A at $10^5$ PFU was inoculated into hamsters ($n = 20$). At 1 dpi, treatment with ensitrelvir at 60 mg/kg twice daily ($n = 5$), nirmatrelvir at 250 mg/kg twice daily ($n = 5$), or molnupiravir at 250 mg/kg twice daily ($n = 5$) was initiated[40,41] and continued until 3 dpi. The remaining hamsters ($n = 5$) were administered 0.5% methyl cellulose. The animals were euthanized at 4 dpi and their nasal turbinates and lungs were collected for the virus titration on VeroE6/TMPRSS2 cells.

## Statistical analysis

GraphPad Prism software version 9.3.0 was used to calculate $P$ values. Virus growth kinetics in vitro, body weight, Penh, and Rpef were compared by using a two-way ANOVA followed by Dunnett's multiple comparisons. Virus titers in hamsters were compared by using a one-way ANOVA followed by Tukey's multiple comparisons. Differences between groups were considered significant for $P$ values < 0.05.

## Reporting summary

Further information on research design is available in the Nature Portfolio Reporting Summary linked to this article.

## Data availability

All data supporting the findings of this study are available within the paper and are provided in the Source data file. There are no restrictions to obtaining access to the primary data. Source data are provided with this paper.

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

## Acknowledgements

We thank Susan Watson for editing the manuscript. We also thank Yuko Sato and Seiya Ozono for technical assistance. This work was supported by the Japan Agency for Medical Research and Development (JP22wm0125002, JP223fa627001, JP22wm0125008, JP22fk0108637, and JP21fk0108552).

## Author contributions

M.K., S.Y. and Y.K. designed the study. M.K., S.Y., S.I., Y.F., Y.H., R.U., T.S. and M.I. performed the experiments. M.K., S.Y., S.I., T.S. and Y.K. analyzed the data and wrote the manuscript. All authors reviewed the manuscript and approved the final version.

## Competing interests

Y.K. has received funding support from Daiichi Sankyo Pharmaceutical, Toyama Chemical, Tauns Laboratories, Inc., Shionogi & Co. LTD, Otsuka Pharmaceutical, and KM Biologics. The other authors declare no competing interests.
