## [Peer Review File · Nature Communications]

Reviewer comments, first round

Reviewer #1 (Remarks to the Author):

Ensitrelvir is a protease inhibitor against the SARS-CoV-2 protease nsp5 approved for treatment of COVID-19 in Japan and on the way to approval in other countries. By passaging SARS-CoV-2 in the presence of increasing concentration of ensitrelvir, the authors selected two resistance mutations against ensitrelvir and a virus variant with both mutations. They studied the degree of resistance also to other antivirals against SARS-CoV2 and analysed the fitness of the virus in cell culture. Mutated viruses were also found to be resistant to ensitrelvir in a hamster model and the E166A mutation was found to attenuate the virus in this model, while the M49L did not. The mutations found have been mentioned elsewhere but not studied in depth in cell culture and in animal models. Especially the findings in hamsters are important, as they show that SARS-CoV-2 can acquire significant resistance to ensitrelvir with a single aa substitution without obviously losing pathogenicity. Luckily, these viruses are still susceptible to other antivirals including nirmatrelvir.

The study is technically sound, the results justify the conclusions and the findings are extremely important. Some minor points should be addressed:

- The authors should briefly discuss that the passaging of wild-type virus in the presence of increasing concentration of an antiviral are gain of function experiments that, if allowed, require additional safety precautions.
- There is a slight decrease in sensitivity of the mutated viruses against remdesivir and molnupiravir. Do the authors have an explanation for this?
- The authors should discuss the limitation that the omicron variant of SARS-CoV-2 has one aa substitution relative to delta and that formally the results must be confirmed for the omicron major protease, as omicron is the currently dominating variant.

Reviewer #2 (Remarks to the Author):

Ensitrelvir is a novel antiviral that recently entered the clinic so it is crucial that its resistance profile is thoroughly characterized. The authors selected ensitrelvir resistant virus in vitro and further characterized this in vitro and in vivo. These results are important for the community to anticipate the potential resistance development in patients.

The methodology is sound and the work is very well written with clear figures and legends. And is properly updated with the literature. We thank the authors to provide such a well worked-out manuscript as it makes the review process more efficient.

However we have two major points of criticism. In our opinion the paper should not be accepted when these are not addressed.

1. The authors studied the replication of the resistant viruses in animals. However they do not evaluate if, during these experiments, the viruses may revert back to wild-type. It might be that the mutant viruses seem to replicate very well but in fact there is a rapid reversion to wild-type and it is the wild-type that is replicating. In my opinion it is essential to confirm the presence of the mutations in the viruses harvested from the animals.
2. At different occasions nirmatrelvir is used in the work, however this compound is always tested in sub-optimal conditions. As the reader may compare the activities of Nirmatrelvir with Ensitrelvir it is important to point this out both in the result and discussion section. Otherwise readers may get the false impression that ensitrelvir is far superior to nirmatrelvir both in vitro and in vivo.
2.a In table 2 nirmatrelvir activity on SARS2 in Vero cells is described (please include cell type in fig legend). However as described in several papers (example Owen et al., Science 374, 1586–1593 (2021)) it is essential to use a PGP inhibitor in VeroE6 cells to get a good estimate of Nirmatrelvir activity. The authors should therefore explicitly mention in the result and discussion

section that in their work the EC50 of nirmatrelvir is ~100x higher than previously reported because of the extreme efflux from the VeroE6 cells.

2.b The authors should mention that in humans nirmatrelvir is dosed with ritonavir to get sufficient exposure. As no ritonavir was used the efficacy of nirmatrelvir in these animal experiments should not be directly compared with the efficacy of ensitrelvir. The authors should indicate this in the result section and discussion when describing the animal experiments.

REVIEWER COMMENTS

Reviewer #1 (Remarks to the Author):

-The authors should briefly discuss that the passaging of wild-type virus in the presence of increasing concentration of an antiviral are gain of function experiments that, if allowed, require additional safety precautions.

First, a similar experiment to obtain virus resistant to nirmatrelvir and ensitrelvir has been published (Jochmans D, et al. The Substitutions L50F, E166A, and L167F in SARS-CoV-2 3CLpro Are Selected by a Protease Inhibitor In Vitro and Confer Resistance To Nirmatrelvir. **mBio** 2023)

Second, a mutant virus with the M49L substitution in nsp5 has already been generated by the same method we applied and shown to have reduced susceptibility to ensitrelvir (https://www.kegg.jp/medicus-bin/japic_med?japic_code=00070668#doc_14). The actual data can be found here (<https://www.pmda.go.jp/files/000249828.pdf>).

Third, clinical isolates possessing the M49L and E166A substitutions are registered in the public database, and it has been reported that these substitutions are involved in reduced susceptibility to ensitrelvir (Moghadasi SA, et al. Transmissible SARS-CoV-2 variants with resistance to clinical protease inhibitors. **Sci Adv** 2023 and Harris R, et al. Rapid resistance profiling of SARS-CoV-2 protease inhibitors. **Res Sq** 2023).

Fourth, antiviral agents with a mechanism of action against SARS-CoV-2 that differs from that of ensitrelvir are available.

Nonetheless, we performed the experiments described in this manuscript by taking the following safety measures:

1. All experiments with SARS-CoV-2 were performed in enhanced biosafety level 3 containment laboratories at the University of Tokyo, which are approved for such use by the Ministry of Agriculture, Forestry, and Fisheries and the Ministry of Health, Labour and Welfare, Japan.
2. All experiments were conducted by PhD-level scientists who are highly experienced in such studies.
3. All individuals working with SARS-CoV-2 have been vaccinated multiple times with COVID-19 vaccines.
4. Staff working in enhanced BSL-3 wear disposable overalls and powered air-purifying respirators.
5. The enhanced BSL-3 facility at the University of Tokyo includes controlled access,

effluent decontamination, negative air-pressure, double-door autoclaves, HEPA-filtered supply and exhaust air, and airtight dampers on ductwork connected to the animal cage isolators and biosafety cabinets.

6. The structure is pressure-decay tested regularly. All personnel complete biosafety and BSL-3 training before participating in BSL-3-level experiments.
7. Refresher training is scheduled on a regular basis.
8. Virus inventory, secured behind two physical barriers, is checked regularly.
9. Virus inventory is submitted once a year to the Ministry of Agriculture, Forestry, and Fisheries and the Ministry of Health, Labour and Welfare, Japan.
10. Procedures in response to laboratory accidents are established.

These points have been described in the main text (Page 4, lines 11–17; Page 6, lines 10–24).

-There is a slight decrease in sensitivity of the mutated viruses against remdesivir and molnupiravir. Do the authors have an explanation for this?

It is not clear why the sensitivity of the mutant viruses to remdesivir and molnupiravir was reduced. This point is now included in the main text (Page 12, lines 24–25).

-The authors should discuss the limitation that the omicron variant of SARS-CoV-2 has one aa substitution relative to delta and that formally the results must be confirmed for the omicron major protease, as omicron is the currently dominating variant.

In response to the reviewer's comment, we have now included this point in the Discussion (Page 16, lines 12–15).

Reviewer #2 (Remarks to the Author):

1. The authors studied the replication of the resistant viruses in animals. However they do not evaluate if, during these experiments, the viruses may revert back to wild-type. It might be that the mutant viruses seem to replicate very well but in fact there is a rapid reversion to wild-type and it is the wild-type that is replicating. In my opinion it is essential to confirm the presence of the mutations in the viruses harvested from the animals.

In response to the reviewer's comment, we performed deep sequencing of the nasal turbinate and lung samples of hamsters infected with wild-type and mutant viruses at 3 and 6 days post-infection. It revealed that additional amino acid substitutions and reversions were not present in most samples, although one nasal turbinate sample from a hamster infected with Nsp5-M49L/E166A at 6 dpi contained the E47K substitution in nsp5 at 17.1%. This E47K substitution may not have had a large effect on viral replication efficiency based to the virus titer. However, further analysis will be required to characterize this substitution because it is located close to the enzyme active site of nsp5. This point is now described in the main text (Page 14, lines 10–18).

2. At different occasions nirmatrelvir is used in the work, however this compound is always tested in sub-optimal conditions. As the reader may compare the activities of Nirmatrelvir with Ensitrelvir it is important to point this out both in the result and discussion section. Otherwise readers may get the false impression that ensitrelvir is far superior to nirmatrelvir both in vitro and in vivo.

2.a In table 2 nirmatrelvir activity on SARS2 in Vero cells is described (please include cell type in fig legend). However as described in several papers (example Owen et al., Science 374, 1586-1593 (2021)) it is essential to use a PGP inhibitor in VeroE6 cells to get a good estimate of Nirmatrelvir activity. The authors should therefore explicitly mention in the result and discussion section that in their work the EC50 of nirmatrelvir is ~100x higher than previously reported because of the extreme efflux from the VeroE6 cells.

In response to the reviewer's comment, the cell type used in Table 1 is now described in the footnote of Table 1. In addition, the point related to the p-glycoprotein inhibitor is explained in both the Results and Discussion sections (Page 12, lines 21–23; Page 17, lines 3–5).

2.b The authors should mention that in humans nirmatrelvir is dosed with ritonavir to get sufficient exposure. As no ritonavir was used the efficacy of nirmatrelvir in these animal experiments should not be directly compared with the efficacy of ensitrelvir. The authors should indicate this in the result section and discussion when describing the animal experiments.

In response to the reviewer's comment, we have mentioned the co-administration of ritonavir in both the Result and Discussion sections (Page 15, lines 15–16; Page 17, lines 3–5).